# Exogenous Lactate Treatment Immediately after Exercise Promotes Glycogen Recovery in Type-II Muscle in Mice

**DOI:** 10.3390/nu16172831

**Published:** 2024-08-24

**Authors:** Taeho Kim, Deunsol Hwang, Sunghwan Kyun, Inkwon Jang, Sung-Woo Kim, Hun-Young Park, Hyejung Hwang, Kiwon Lim, Jisu Kim

**Affiliations:** 1Laboratory of Exercise and Nutrition, Department of Sports Medicine and Science in Graduate School, Konkuk University, Seoul 05029, Republic of Korea; qwer92224@konkuk.ac.kr (T.K.); hds49@konkuk.ac.kr (D.H.); y10345@konkuk.ac.kr (S.K.); inkwon555@konkuk.ac.kr (I.J.); kswrha@konkuk.ac.kr (S.-W.K.); parkhy1980@konkuk.ac.kr (H.-Y.P.); hfilm@konkuk.ac.kr (H.H.); exercise@konkuk.ac.kr (K.L.); 2Physical Activity and Performance Institute (PAPI), Konkuk University, Seoul 05029, Republic of Korea; 3Department of Physical Education, Konkuk University, Seoul 05029, Republic of Korea

**Keywords:** exogenous lactate, glycogen recovery, glycogen synthesis, supercompensation, energy supplement

## Abstract

Recent studies suggest that lactate intake has a positive effect on glycogen recovery after exercise. However, it is important to verify the effect of lactate supplementation alone and the timing of glycogen recovery. Therefore, in this study, we aimed to examine the effect of lactate supplementation immediately after exercise on glycogen recovery in mice liver and skeletal muscle at 1, 3, and 5 h after exercise. Mice were randomly divided into the sedentary, exercise-only, lactate, and saline-treated groups. mRNA expression and activation of glycogen synthesis and lactate transport-related factors in the liver and skeletal muscle were assessed using real-time polymerase chain reaction. Skeletal muscle glycogen concentration showed an increasing trend in the lactate group compared with that in the control group at 3 and 5 h after post-supplementation. Additionally, exogenous lactate supplementation significantly increased the expression of core glycogen synthesis enzymes, lactate transporters, and pyruvate dehydrogenase E1 alpha 1 in the skeletal muscles. Conversely, glycogen synthesis, lactate transport, and glycogen oxidation to acetyl-CoA were not significantly affected in the liver by exogenous lactate supplementation. Overall, these results suggest that post-exercise lactate supplement enables glycogen synthesis and recovery in skeletal muscles.

## 1. Introduction

Glycogen supplies glucose to cells during muscle contraction and is the most efficiently used energy source [1,2,3,4]. It is stored in the liver and muscles at an average amount of approximately 500 g in a 70 kg adult male. Glycogen depletion in the body can induce a decrease in exercise capacity and fatigue [5,6,7,8]; therefore, athletes are placed on carbohydrate-centered menus to facilitate glycogen recovery after exercise. Additionally, glycogen- or carbohydrate-loading strategies are adopted to increase prematch carbohydrate intake and match-day glycogen stores [2,9,10,11]. However, these strategies have side effects such as gastrointestinal disorders and indigestion [2,7,12]. Moreover, most commercially available nutrients and exercise supplements contain scientifically untested plant root and fruit extracts [13], which might negatively impact athletic performance [14,15,16]. Therefore, studies are underway to identify new exercise aids and strategies and potential energy-boosting substances produced in the body [2,9,13,15].

Among the substances produced in the body, the gluconeogenesis precursor lactate has attracted extensive research attention, leading to an improved understanding of its role [17,18,19,20]. Cori et al. found that lactate produced in muscles during anaerobic glycolysis is transported to the liver, where it is converted back into glucose. This glucose is then reused by the muscles as an energy source, a process now known as the “Cori cycle” [21]. This discovery advanced our understanding of lactate. Furthermore, the identification of monocarboxylate transporters (MCTs) provided critical insight into the mechanism by which lactate is transported in and out of cells, further elucidating the complex interplay between metabolic processes in different tissues [18]. Additionally, lactate can be transported to energy-deficient tissues to serve as an energy source by monocarboxylate transporters (MCTs) [19,22,23].

Animal studies (rat) have shown that exogenous lactate treatment reportedly increases glycogen synthesis in type-II muscles [24]. Additionally, perfusion with lactate increases glycogen concentration in the gastrocnemius muscle of rats [25], and intraperitoneal administration of lactate after exercise significantly affects glycogen recovery in type-II muscles in mice [5]. However, it is difficult to conclude that lactate alone is responsible for these effects, as most previous studies used combination treatments, such as lactate and glucose. For instance, combined oral treatment with lactate and caffeine significantly affected glycogen synthesis and storage in the liver [26]. Overall, these studies suggest that exogenous treatment with lactate positively affects glycogen synthesis and storage. To our knowledge, no previous studies have definitively confirmed the effects of a single administration of exogenous lactate. Furthermore, our study design, which evaluates the effects of exogenous lactate intake up to 5 h post-exercise and investigates the timing of intake, is expected to provide valuable insights into the duration of lactate’s effects. Based on these findings, it is important to elucidate the effects of exogenous lactate intake on glycogen synthesis in the liver and muscle and ascertain changes in glycogen concentration at different time points following lactate supplement. Our research team hypothesized that even a single acute treatment with exogenous lactate would positively impact glycogen recovery. Therefore, in this study, we aimed to examine the effects of exogenous lactate intake immediately after exercise on glycogen recovery at 1, 3, and 5 h post-administration and elucidate key factors of glycogen synthesis during recovery. Finally, through this study, we are expected to contribute to identifying the flow of recovery over time and regulating effective lactate intake timing and exercise time. Additionally, if the acute intake of exogenous lactate after exercise has a positive effect on energy recovery, it suggests the potential role of lactate as a catalytic substance aiding recovery.

## 2. Materials and Methods

### 2.1. Animal Care

Eight-week-old male ICR mice (body weight 36.5 ± 2.1) were purchased from orient Bio (Seongnamsi, Gyeonggido, Republic of Korea) and randomly grouped in standard laboratory animal cages (*n* = 64). The animals were housed under controlled conditions (12:12 h light:dark cycle; 50% humidity; temperature, 23 ± 2 °C) and had unlimited access to standard rodent diet (rodent-type diet 5L79, Orient Bio, Seongnam, Gyeonggido, Republic of Korea) and water.

### 2.2. Experimental Design

The experiment was designed to determine the effect of an acute (one-time) treatment of exogenous lactate. The mice were randomly divided into eight groups (Figure 1): sedentary group without any treatment (CON; *n* = 8), exercise-only group (EXE; *n* = 8), and lactate- and saline-treated group with oral administration of lactate and saline, respectively, immediately after exercise (LAC; *n* = 24; SAL; *n*= 24), which were further subdivided by 1, 3, and 5 h time points (LAC-1, LAC-3, LAC-5; SAL-1, SAL-3, SAL-5; *n* = 8 mice/group). All mice were acclimated to the laboratory environment for one week. Exercise adaptation training was conducted on an animal treadmill for 3 days. The exercise intensity in this experiment was set at VO2max80% to deplete glycogen in the skeletal muscles and liver. A detailed exercise protocol is provided in the following paragraphs. And lactate and saline water were orally administered to the LAC and SAL groups, respectively, immediately after the treadmill exercise at 3 g/kg (sodium L-lactate, L7022, Sigma) [26,27,28,29]. The mice were then sacrificed at the resting times (1, 3, and 5 h) determined for each group.

### 2.3. Exercise Protocol

All mice underwent exercise adaptation training conducted on an animal treadmill for 3 days (speed, 10–15 m/min; duration, 20 min; incline, 0°). The exercise intensity in this experiment was set at VO2max80%, according to the VO2max protocol [22,25,26,27], to deplete glycogen in the skeletal muscles and liver (speed, 28 m/min; duration, 50 min; incline, 15°). The main exercise protocol was as follows: a 10 min warm-up was performed (speed, 4–8 m/min; incline, 0°), with the speed gradually increased to allow for exercise adaptation. The main exercise session lasted for 50 min (speed, 28 m/min; incline, 15°). Most of the mice completed the 50 min exercise session, but four mice were judged to be in a state of exhaustion (All-out) in the latter half of the exercise time, and the exercise was terminated.

### 2.4. Blood Analysis

Blood samples collected from the mice via the tail vein before sacrifice were used for blood glucose (ACCU CHEK Performa Glucometer, Roche Diagnostics, Penzberg, Germany) and blood lactate (Lactate Pro2, LT-1730, ARKRAY, Kyoto, Japan) analysis using their respective kits.

### 2.5. Glycogen Concentration Analysis

To clarify the glycogen recovery effect of exogenous lactate treatment, we measured glycogen concentrations in the liver and plantaris muscle of the mice. Previous studies indicated that the plantaris muscle, which is primarily used during high-intensity exercise, showed the best glycogen recovery pattern [5,24]. Therefore, we chose to use the plantaris muscle for our measurements. Briefly, the glycogen contents of liver tissue and plantaris muscle were measured using a Glycogen Assay Kit (K646-100; BioVision, Waltham, MA, USA) according to the manufacturer’s instructions.

### 2.6. Real-Time Reverse Transcription Polymerase Chain Reaction (qRT-PCR)

Real-time RT-PCR analysis was performed to confirm the mRNA expression and activation of glycogen synthesis and lactate transport-related factors in liver and skeletal muscle. Briefly, total RNA was extracted from the liver and plantaris muscle (type-II skeletal muscle) using QIAzol Lysis Reagent (79306; Qiagen, Hilden, Germany). cDNA was reverse-transcribed from the RNA samples using cDNA Synthesis Platinum Master Mix (reaction condition: annealing for 5 min at 25 °C, extension for 50 min at 42 °C, RT inactivation for 15 min at 70 °C, and final extension at 4 °C). Finally, qRT-PCR was performed on the Quantstudio1 qRT-PCR System (Thermo Fisher, Gangnam, Republic of Korea) using Powerup SYBR Green Master Mix and specific primers (20 μL). The PCR conditions were as follows: 40 cycles of denaturation for 10 min at 95 °C, annealing for 1 min at 59.1 °C, extension for 15 s at 95 °C, and final extension at 4 °C. The expression levels of the target genes (14 genes) were normalized to that of the housekeeping gene glyceraldehyde 3- phosphate dehydrogenase (GAPDH) and calculated using the ∆∆C_T_ method (Figure 2). The sequences of the target genes are shown in Table 1.

### 2.7. Statistical Analysis

All data were analyzed using IBM SPSS Statistics software (version 28.0; Armonk, NY, USA), and graphs were constructed using GraphPad Prism software (version 9.0). All data were examined for normality of distribution using the Shapiro–Wilk or Kolmogorov–Smirnov test to verify normality of all data. Significant differences between two or more groups at each time point were determined using two-way repeated analysis of variance (ANOVA). Post hoc tests were performed using one-way repeated ANOVA and independent *t*-tests. Means were considered statistically significant at *p* < 0.05. Data are presented as mean ± standard error of the mean.

## 3. Results

### 3.1. Blood Lactate and Blood Glucose Level

Blood analysis was performed to examine changes in blood lactate and glucose levels following exogenous lactate treatment (Figure 3). The blood lactate concentration was significantly higher (*p* = 0.017) in the EXE group (7.9 ± 4.2 mmol/L) than in the CON group (3.2 ± 0.5 mmol/L). Although no significant differences were observed between the LAC and SAL groups, there was a trend toward higher blood lactate values in the LAC-3h group (10.8 ± 8.6 mmol/L) than in the SAL-3h group (4.3 ± 2.2 mmol/L) (Figure 3A). Conversely, the blood glucose level was significantly lower (*p* < 0.001) in the EXE group (88.2 ± 34.3 mg/dL) than in the CON group (168.1 ± 17.1 mg/dL). However, no significant difference in blood glucose levels between the LAC and SAL groups was observed at any time point, with a gradual decrease in blood glucose levels observed over time (Figure 3B). The normality of the blood analysis data was assessed using the Shapiro–Wilk or Kolmogorov–Smirnov test. The results indicated that the data were normally distributed (*p* > 0.05). Therefore, parametric tests were applied in further analyses.

### 3.2. Muscle Weight and Muscle Glycogen Concentration

Next, the plantaris muscle weight and glycogen concentration were measured after exercise to confirm the effect of exogenous lactate treatment on glycogen recovery in skeletal muscle (Figure 4). No significant difference in muscle weight was observed between the EXE (0.50 ± 0.05 mg/g BW) and CON (0.49 ± 0.04 mg/g BW) groups; however, the SAL-3h group (0.49 ± 0.05 mg/g BW) had a significantly higher muscle weight (*p* = 0.025) than the LAC-3h group (0.43 ± 0.03 mg/g BW) (Figure 4A). The muscle glycogen concentration was significantly lower in the EXE group (0.006 ± 0.002 μg/μL) than in the CON group (0.045 ± 0.011 μg/μL) (*p* < 0.001). In contrast, the SAL-1 group (0.106 ± 0.033 μg/μL) had a significantly higher (*p* = 0.023) muscle glycogen concentration than the LAC-1 group (0.062 ± 0.035 μg/μL) (Figure 4B). However, the SAL group showed a gradual decrease in the glycogen concentration after 3 h of exercise (Figure 4D). The highest muscle glycogen concentration was measured in the LAC-3h group (0.109 ± 0.028 μg/μL) (Figure 4C). Additionally, the LAC-5h group (0.067 ± 0.026 μg/μL) showed an increasing trend compared with the SAL-5h group (0.047 ± 0.012 μg/μL) (*p* = 0.065). Overall, lactate ingestion after exercise tended to compensate for glycogen in the plantaris muscle after 3 h; however, no direct effect on muscle weight was confirmed. The normality of the muscle weight and glycogen concentration analysis data was assessed using the Shapiro–Wilk or Kolmogorov–Smirnov test. The results indicated that the data were normally distributed (*p* > 0.05). Therefore, parametric tests were applied in further analyses.

### 3.3. Glycogen Synthase and MCT-1,4 Gene Expression in Muscle

We next examined the expression of core enzymes, *Hexokinase2* (HK2), *Glycogen synthase1* (GYS1), and *Phosphoglucomautase1* (PGM1), involved in glycogen synthesis in the skeletal muscle using qRT-PCR to elucidate the molecular mechanism of exogenous lactate treatment on glycogen recovery after exercise (Figure 5). No significant differences in the gene expression levels of HK2, GYS1, and PGM1 were observed between the CON and EXE groups. However, HK2 gene expression was significantly higher in the LAC group than in the SAL group at 1 h (*p* = 0.012), 3 h (*p* = 0.024), and 5 h (*p* = 0.019) post-exercise (Figure 5A). Similarly, GYS1 gene expression was significantly higher in the LAC group than in the SAL group at 1 h (*p* = 0.017), 3 h (*p* = 0.009), and 5 h (*p* = 0.015) post-exercise (Figure 5B). Moreover, the LAC group showed significantly higher PGM1 gene expression than the SAL group at 1 h (*p* < 0.001) and 3 h (*p* = 0.007) post-exercise (Figure 5C). Additionally, to confirm that exogenous lactate treatment was used as an energy source in muscle cells, we examined the expression of *Pyruvate dehydrogenase E1 alpha 1* (PDHA1). Consistent with the results of glycogen synthesis enzymes, no significant difference in PDHA1 gene expression was observed between the CON and EXE groups. However, PDHA1 gene expression was significantly higher in the LAC group than in the SAL group at 1 h (*p* = 0.018) and 3 h (*p* = 0.01) after exercise. No significant difference (*p* = 0.06) in PDHA1 gene expression was observed between the LAC and SAL groups at 5 h after exercise (Figure 5D). Furthermore, we examined the expression of the lactate transporters, MCT1 and MCT4, to confirm lactate transportation in muscle cells. No significant difference in MCT1 and MCT4 gene expression levels was observed between the CON and EXE groups. However, MCT1 gene expression was significantly higher in the LAC group than in the SAL group at 1 h (*p* = 0.004), 3 h (*p* = 0.009), and 5 h (*p* = 0.023) after exercise (Figure 5E). Similarly, MCT4 gene expression was significantly higher in the LAC group than in the SAL group at 1 h (*p* = 0.004) and 3 h (*p* = 0.034) after expression. However, no significant difference (*p* = 0.328) in MCT4 gene expression was observed between the LAC and SAL groups at 5 h after exercise (Figure 5F). Overall, increased expression of MCT1, MCT4, and glycogen synthase was observed in the LAC group. The normality of the muscle qRT-PCR analysis data was assessed using the Shapiro–Wilk or Kolmogorov–Smirnov test. The results indicated that the data were normally distributed (*p* > 0.05). Therefore, parametric tests were applied in further analyses.

### 3.4. Liver Weight and Liver Glycogen Concentration

We examined the effect of exogenous lactate treatment after exercise on the liver weight and glycogen content because the liver is the major glycogen store in the body. The liver weight was significantly lower (*p* = 0.001) in the EXE group (45.3 ± 2.1 mg/g BW) than in the CON group (51.8 ± 3.8 mg/g BW) (Figure 6A); however, no significant difference was observed between the LAC and SAL groups. Similarly, the liver glycogen concentration was significantly lower (*p* < 0.001) in the EXE group (0.234 ± 0.122 μg/μL) than in the CON group (0.957 ± 0.187 μg/μL) (Figure 6B). No significant difference in the liver glycogen concentration was observed between the LAC and SAL groups. However, the liver glycogen concentration was significantly higher in the LAC-3h (0.604 ± 0.151 μg/μL; *p* < 0.001), LAC-5h (0.569 ± 0.234 μg/μL; *p* = 0.015), SAL-3h (0.597 ± 0.314 μg/μL; *p* = 0.029), and SAL-5h (0.576 ± 0.323 μg/μL; *p* = 0.018) groups than in the LAC-1 and SAL-1 groups, respectively (Figure 6B). The results in the liver show that exogenous lactate intake after exercise does not alter the liver weight or glycogen concentration. The normality of the liver weight and glycogen concentration analysis data was assessed using the Shapiro–Wilk or Kolmogorov–Smirnov test. The results indicated that the data were normally distributed (*p* > 0.05). Therefore, parametric tests were applied in further analyses.

### 3.5. Gluconeogenesis Gene Expression in Liver

Next, we examined the expression of *Pyruvate carboxylase* (PC), *Phosphoenolpyruvate carboxykinase 1* (PCK1), *Fructose bisphosphatase 1* (FBP1), and *Glucose-6-phosphatase* (G6Pase) in the liver using qRT-PCR to confirm the effect of exogenous lactate ingestion on gluconeogenesis (Figure 7). No significant differences in the gene expression levels of PC, PCK1, FBP1, and G6Pase were observed between the CON and EXE groups (Figure 7A–D). However, PCK1 gene expression was significantly higher in the SAL-1h group than in the LAC-1h group (*p* = 0.042). Additionally, PCK1 gene expression was significantly lower in the LAC-3h (*p* = 0.022), LAC-5h (*p* = 0.035), SAL-3h (*p* = 0.025), and SAL-5h (*p* = 0.025), groups than in the EXE group (Figure 7B). Similarly, although no significant difference in FBP1 gene expression was observed between the LAC and SAL groups at any time point, FBP1 gene expression was significantly higher in the SAL-1h group than in the EXE group (*p* = 0.003) (Figure 7C). No significant difference in G6Pase gene expression was observed between the LAC and SAL groups at any time point; however, G6Pase gene expression was significantly lower in the LAC and SAL groups than in the EXE group at 3 h (*p* < 0.001; *p* < 0.001, respectively) and 5 h (*p* = 0.002; *p* < 0.001, respectively) after exercise (Figure 7D). These results suggest that exogenous lactate intake after exercise does not alter liver gluconeogenesis. The normality of the liver qRT-PCR analysis data was assessed using the Shapiro–Wilk or Kolmogorov–Smirnov test. The results indicated that the data were normally distributed (*p* > 0.05). Therefore, parametric tests were applied in further analyses.

### 3.6. Glycogen Synthesis and MCT-1,4 Gene Expression in Liver

We examined the gene expression of *Glucokinase* (GK), *Glycogen synthase 2* (GYS2), MCT-1, and MCT-4 in mice livers to confirm whether exogenous lactate treatment affected lactate transport and glycogen synthesis in the liver (Figure 7E–H). No significant differences in GK, GYS2, MCT-1, and MCT-4 gene expression levels were observed between the CON and EXE groups. Similarly, GK gene expression was not significantly different between the LAC and SAL groups at any time point; however, it was significantly higher in the LAC-1h (*p* = 0.006) and SAL-1h (*p* = 0.035) groups than in the EXE group. Additionally, the within-group comparison showed that GK gene expression was significantly lower in the LAC-5h group than in the LAC-1h group (*p* = 0.026) and significantly lower in the SAL-5h group than in the SAL-1h group (*p* = 0.009) (Figure 7E). Although no significant difference in GYS2 gene expression was observed between the LAC and SAL groups, GYS2 gene expression was significantly lower in the LAC-3h (*p* = 0.007) and LAC-5h (*p* = 0.03) groups than in the EXE group. Moreover, the within-group comparison showed that GYS2 gene expression was significantly lower in the LAC-3h (*p* = 0.033) and LAC-5h (*p* = 0.044) groups than in the LAC-1h group at the respective time points (Figure 7F). Furthermore, no significant differences in MCT-1 and MCT-4 gene expression levels were observed between the LAC and SAL groups at any time point (Figure 7G,H). The normality of the liver qRT-PCR analysis data was assessed using the Shapiro–Wilk or Kolmogorov–Smirnov test. The results indicated that the data were normally distributed (*p* > 0.05). Therefore, parametric tests were applied in further analyses.

## 4. Discussion

Lactate is recognized as a precursor in gluconeogenesis [17,18,20,30]. We hypothesized that exogenous lactate supplementation would enhance glycogen recovery post-exercise. Specifically, we posited that ingesting exogenous lactate immediately after exercise would augment glycogen recovery for up to 5 h. To evaluate this hypothesis, we examined the impact of a single lactate treatment on blood lactate, blood glucose, glycogen concentration, tissue weight, gluconeogenesis, and glycogen synthesis enzyme expression. Measurements were taken via tail blood sampling, liver, and skeletal muscle analysis at 1, 3, and 5 h post-treatment. First, we analyzed the effect of orally administered exogenous lactate on blood lactate concentration using tail blood sampling. Immediately after exercise, exogenous lactate administration tended to increase the mean blood lactate concentration compared to the saline (SAL) group 3 h post-administration. However, this increase was not statistically significant. Although the orally administered lactate dose of 3 g/kg was relatively high, the lack of a significant difference between the groups suggests that the lactate was rapidly utilized by tissues requiring an energy source after exercise. Notably, the skeletal muscle glycogen concentration was higher in the LAC-5h group than in the SAL-5h group, indicating a trend toward recovery (Figure 4B), which was consistent with the results of a previous study on the effect of chronic lactate intake after chronic exercise (speed, 25 m/min; duration, 40min; 6 day/week for 3 wk) [22]. However, unlike that in the chronic exercise experiment, the glycogen concentration of the skeletal muscle was improved by acute lactate treatment in the present study. Interestingly, the LAC-3h group, which had the highest skeletal muscle glycogen concentration, had the lowest skeletal muscle weight (Figure 4A). A review article on muscle glycogen and body water identified a correlation between muscle glycogen content and weight gain in humans [31]. However, Sherman et al. found no consistent relationship between the water volume and glycogen content in rodent skeletal muscle, highlighting potential species-specific differences in the relationship between glycogen storage and water retention [32]. Therefore, it could be concluded that the glycogen concentration increases even if muscle weight decreases. Although exogenous lactate was administered as an acute treatment, our results showed an alteration in skeletal muscle glycogen concentration.

Exogenous lactate increased the expression of glycogen synthase in skeletal muscle. An analysis of the core enzymes involved in glycogen synthesis in muscles showed that *Hexokinase2* (HK2) increases glucose phosphorylation and affects glycogen recovery [33,34]. Additionally, an increase in the expression of HK2, often referred to as the “guardian of mitochondria” [35], has been linked to improved muscle endurance. Fueger et al. observed that higher levels of hexokinase (HK) protein content were specifically associated with enhanced muscle endurance in their study [36]. In contrast, a previous study used tissues from experimental animals to culture muscle tissue in the laboratory with various concentrations of lactate and observed that this inhibited HK2 activity [37]. However, exogenous lactate supplementation significantly increased the gene expression of HK2, a glycogen synthesis factor, in the plantaris muscle of mice at 1, 3, and 5 h post-exercise in the present study (Figure 5A). To the best of our knowledge, this is the first study to show that acute exogenous lactate supplementation after exercise can increase HK2 gene expression, which can affect glycogen recovery. Additionally, *Glycogen synthase1* (GYS1), another core enzyme involved in glycogen synthesis, affects the amount of stored glycogen depending on the expression level [38]. In the present study, exogenous lactate supplementation increased GYS1 gene expression at all time points (1, 3, and 5 h) (Figure 5B). Although GYS1 gene expression levels were inconsistent with changes in the glycogen concentration based on the mean glycogen concentration, high gene expression of GYS1, particularly at 3 h after exercise, appeared to directly increase glycogen concentration [38,39]. Phosphoglucomutase1 (PGM1) is a catalytic enzyme involved in both glycogen synthesis and degradation and has recently attracted attention as a key regulator of glycogen synthesis and metabolism [3]. Compared with the levels in the SAL group, exogenous lactate supplementation increased PGM1 gene expression at both 1 and 3 h after exercise (Figure 5C). However, no significant difference in PGM1 gene expression was observed between the LAC-5h and SAL-5h groups. Overall, PGM1 gene expression was reduced to CON levels at 5 h, indicating that exogenous lactate supplementation alone contributed to energy synthesis up to 3 h. *Pyruvate dehydrogenase α1* (PDHA1) is one of the three enzymes of the pyruvate dehydrogenase complex and is a key regulatory enzyme in carbohydrate oxidation [40,41]. A high expression of the PDHA1 gene was observed in the LAC group at 1 and 3 h after exogenous lactate treatment. Although no significant difference was observed between the LAC and SAL groups at 5 h, *pdh-α1* showed an increasing trend in the LAC-5h group compared with that in the SAL-5h group (Figure 5D). Reportedly, exercise increases PDHA1 expression [42] and maintains a fairly high PDHA1 activity for 120 min in the liver treated with exogenous lactate [26]. Similarly, PDHA1 gene expression in skeletal muscle increased at the mRNA level following exogenous lactate supplementation in the present study, which improved the energy-generation capacity. Additionally, the gene expression of *Pyruvate dehydrogenase kinase 4*, an enzyme that suppresses PDHA1, was significantly lower in the group treated with exogenous lactate before exercise, indicating increased carbohydrate oxidation [27]. Based on these results, it can be concluded that exogenous lactate supplementation after exercise promotes energy production and glycogen synthesis. *Monocarboxylate transporter 1,4* (MCT1,4) transports lactate between skeletal muscle cells, plays a role in transferring lactate from the blood to the mitochondria, and induces lactate oxidation [18,19]. In the present study, exogenous lactate treatment increased MCT1 gene expression in the muscles of mice (Figure 5E), which was consistent with the results of a previous long-term study [22]. Conversely, MCT4 transports lactate from skeletal muscle into the blood [19,43] and is usually not well expressed in type-II skeletal muscle, which is attributed to the characteristics of white muscles [44]. A previous study showed that exercise alone did not increase MCT4 gene expression in type-II muscles [45]. However, exogenous lactate supplementation significantly increased *MCT4* gene expression in type-II muscle for up to 3 h compared with that in the SAL group in the present study (Figure 5F). We speculate that lactate is primarily utilized in type-II skeletal muscle after high-intensity exercise.

Furthermore, we examined the effect of exogenous lactate supplementation on liver weight and glycogen concentration. No significant differences in liver weight and glycogen concentration were observed between the LAC and SAL groups (Figure 6). Although no direct effect on glycogen concentration was observed, we examined the effect of lactate on the expression of four enzymes involved in gluconeogenesis because lactate is a known precursor of gluconeogenesis [46,47,48,49]. *Pyruvate carboxylase* (PC), which catalyzes the reaction of pyruvate with oxaloacetate, plays an important role in gluconeogenesis [50,51]; however, exogenous lactate supplementation did not significantly affect PC gene expression in the liver tissue (Figure 7A). Therefore, we examined the mRNA level of *Phosphoenolpyruvate carboxykinase1* (PCK1), an enzyme that regulates gluconeogenesis [52,53]. Notably, PCK1 gene expression was significantly lower in the LAC-1h group than in the SAL-1h group (Figure 7B). Previously reported research results have shown that normal blood glucose levels can be maintained even when PCK1 is inhibited [54]. Compared with the results of studies, the expression of the PCK1 gene was lower in the LAC group than in the EXE group, confirming that lactate was not used as a precursor during gluconeogenesis. Similarly, *Glucose-6-phosphatase* (G6Pase) gene expression was significantly lower in the LAC group than in the EXE group, and there was no significant difference in *Fructose bisphosphatase1* (FBP1) expression between the LAC and EXE groups, confirming that exogenous lactate has no effect in the liver (Figure 7C,D). Additionally, the gene expression of *Glucokinase* (GK) and *Glycogen synthase2* (GYS2), which are key enzymes for glycogen synthesis in the liver, were examined. Exogenous lactate supplementation did not significantly affect GK and GYS2 gene expression in the liver but rather decreased their expression over time (Figure 7E,F). This was contrary to the results observed in skeletal muscle. Several studies have confirmed that skeletal muscle is dependent on energy utilization, and glycogen is preferentially synthesized in skeletal muscle in laboratory animals under high-intensity exercise [55,56,57,58]. Overall, these results indicate that post-exercise treatment with exogenous lactate promotes glycogen synthesis preferentially in skeletal muscle rather than in the liver.

This study is the first to investigate the acute effects of exogenous lactate intake after exercise on glycogen recovery in the liver and skeletal muscle. Conclusively, exogenous lactate supplementation had a positive effect on glycogen synthesis in skeletal muscles after exercise, with high glycogen levels for up to 5 h. Consistent with the muscle glycogen concentration, HK2, GYS1, and PGM1, which are core enzymes for glycogen synthesis in the skeletal muscle, were significantly higher in the LAC group for up to 5 h. In contrast, exogenous lactate intake did not affect the expression of key enzymes involved in gluconeogenesis in the liver, which is attributed to the increased dependence on skeletal muscles for energy synthesis after high-intensity exercise [55,56,57].

The limitation of this study is that it analyzed only the glycogen synthesis pathway and gluconeogenesis pathway, making it difficult to clearly identify the effects of exogenous lactate in the liver. Generally, hepatic gluconeogenesis is known to be a prioritized process to regulate decreased blood glucose levels [59,60]. The results of this study showed that the blood glucose levels of the mice remained within the normal range of 100–140 mg/dL, even after fasting for 5 h following exercise [61] (Figure 3B). Therefore, this suggests that the catabolic pathway is activated in the liver rather than the synthesis pathway. Interestingly, in our previous study, we measured changes in substrate utilization post-exercise exogenous lactate treatment and found that the respiratory exchange ratio (RER) was measured to be low [62]. Based on these findings, exogenous lactate is considered a substance that inhibits carbohydrate utilization after exercise and promotes the recovery of glycogen, the primary energy source. The findings of this study indicate that exogenous lactate promotes glycogen recovery even when administered acutely. This suggests that exogenous lactate may play a beneficial role as a substance that catalyzes energy recovery in sports requiring multiple competitions in a single day, such as in tournament formats. In addition, the group treated with exogenous lactate showed a tendency for glycogen recovery in skeletal muscle, necessitating further research to verify how this recovered glycogen affects actual exercise performance. Therefore, further research is needed to verify the effects of lactate on not only energy recovery but also exercise performance. Furthermore, detailed experiments should be conducted to assess the catalytic role of lactate in enhancing energy recovery and exercise performance by confirming the effects of lactate treatment without dietary restrictions.

## 5. Conclusions

Although this was an acute study where exogenous lactate was administered immediately after exercise, the results demonstrated a significant tendency for increased glycogen recovery and enhanced expression of enzymes involved in the synthesis pathway in skeletal muscles. However, it was challenging to confirm the effect of lactate on the liver. These findings suggest that exogenous lactate can be used as a substance that catalyzes energy recovery in muscle. Moreover, it is essential to investigate the impact of short-term glycogen recovery on exercise performance. Therefore, future studies should evaluate the effects of exogenous lactate intake on exercise performance metrics.

## Figures and Tables

**Figure 1 nutrients-16-02831-f001:**
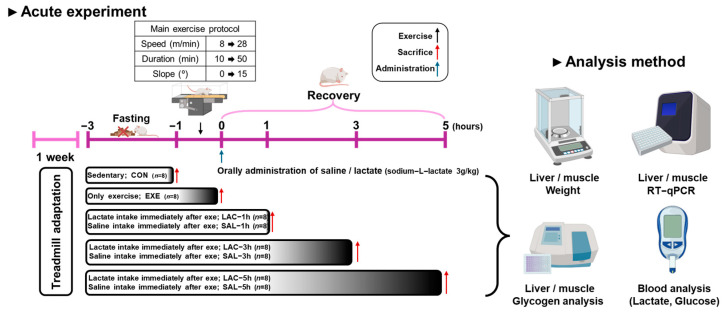
Schematic representation of the experimental procedure. Experiments were initiated with eight-week-old male ICR mice (*n* = 64). This study was designed to observe the effects of exogenous lactate treatment at 1, 3, and 5 h post-exercise. (*n* = 8) CON; sedentary without exercise and lactate, EXE; exercise without lactate, LAC−1h, and SAL−1h; intake lactate or saline immediately after exercise and rest for 1 h. LAC−3h and SAL−3h; intake lactate or saline immediately after exercise and rest for 3 h. LAC−5h and SAL−5h; intake lactate or saline immediately after exercise and rest for 5 h.

**Figure 2 nutrients-16-02831-f002:**
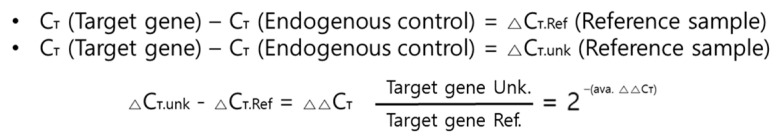
Gene expressions were determined using the ΔΔCt method.

**Figure 3 nutrients-16-02831-f003:**
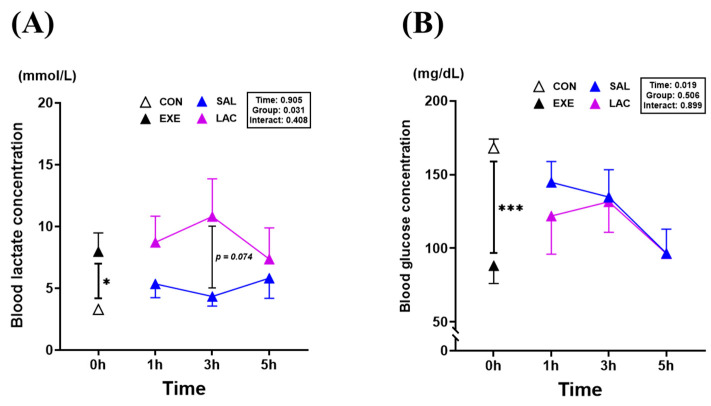
Effects of exogenous lactate treatment on blood lactate and glucose levels. (**A**) Blood lactate concentration, (**B**) Blood glucose concentration values represent the mean ± standard error of the mean (*n* = 8). * *p* < 0.05, *** *p* < 0.001.

**Figure 4 nutrients-16-02831-f004:**
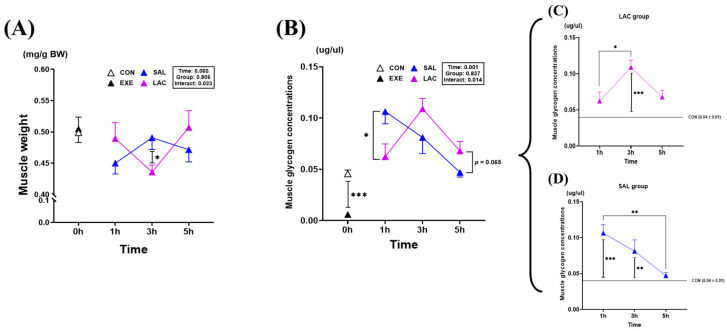
Glycogen concentration and weight in the plantaris muscle. (**A**) Plantaris muscle weight, (**B**) Plantaris muscle glycogen concentration, (**C**) Plantaris muscle glycogen concentration in LAC group, (**D**) Plantaris muscle glycogen concentration in SAL group. Values represent the mean ± standard error of the mean (*n* = 8). * *p* < 0.05, ** *p* < 0.01, *** *p* < 0.001.

**Figure 5 nutrients-16-02831-f005:**
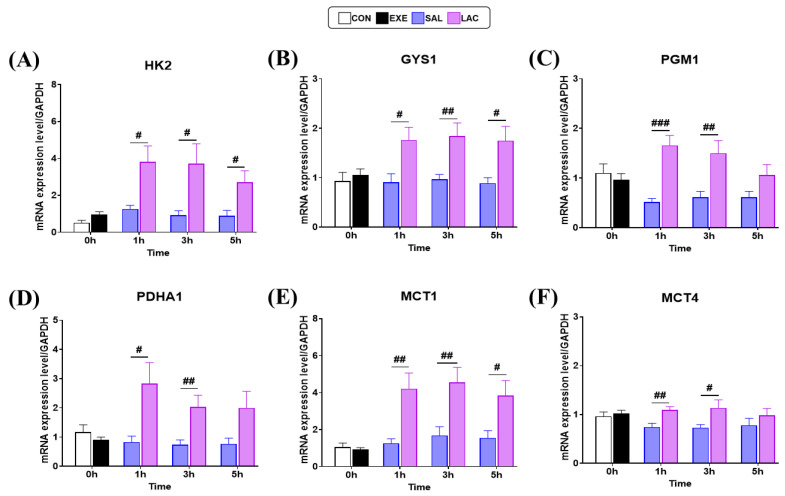
Gene expression of glycogen synthesis and energy metabolism-related mRNAs in the plantaris muscle: (**A**–**F**) qRT-PCR of HK2, GYS1, PGM1, PDHA1, MCT1, and MCT4 mRNA levels, respectively; HK2, *Hexokinase 2*; GYS1, *Glycogen synthase 1*; PGM1, *Phosphoglucomutase 1*; PDHA1, *Pyruvate dehydrogenase α1*; MCT1, *Monocarboxylate transporter 1*; Mct-4, *Monocarboxylate transporter 4*; GAPDH (*Glyceraldehyde 3-phosphate dehydrogenase*) was used for the normalization of the target mRNA expression. Values represent the mean ± standard error of the mean (*n* = 8). # *p* < 0.05, ## *p* < 0.01, ### *p* < 0.001.

**Figure 6 nutrients-16-02831-f006:**
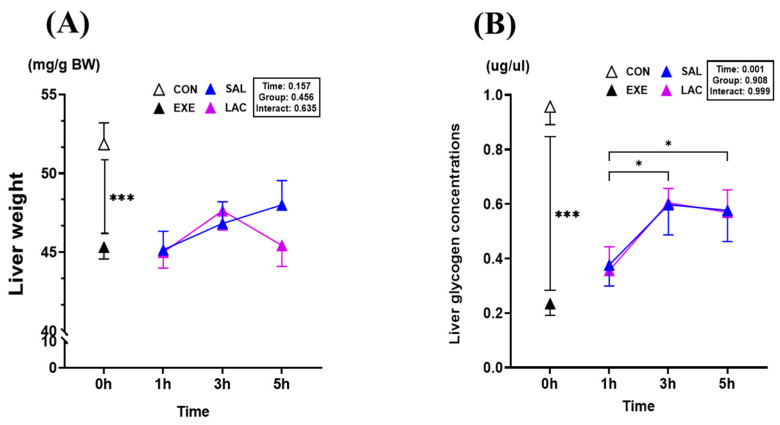
Glycogen concentration and weight in the liver. (**A**) Liver weight, (**B**) Liver glycogen concentration values represent the mean ± standard error of the mean (*n* = 8). * *p* < 0.05, *** *p* < 0.001.

**Figure 7 nutrients-16-02831-f007:**
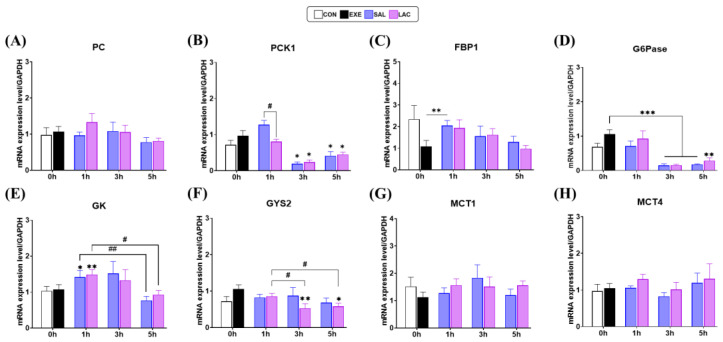
Gene expression of gluconeogenesis- and glycogen synthase-related mRNAs in the liver: (**A**–**H**) qRT-PCR of PC, PCK1, FBP1, G6Pase, GK, GYS2, MCT1, and MCT4 mRNA levels, respectively; PC, *Pyruvate carboxylase*; PCK1, *Phosphoenolpyruvate carboxykinase*; FBP1, *Fructose bisphosphatase 1*; G6Pase, *Glucose-6-phosphatase*; GK, *Glucokinase;* GYS2, *Glycogen synthase 2*; MCT1, *Monocarboxylate transporter 1*; MCT4, *Monocarboxylate transporter 4*; GAPDH (*Glyceraldehyde 3-phosphate dehydrogenase*) was used for the normalization of the target mRNA expression. Values represent the mean ± standard error of the mean (*n* = 8). # *p* < 0.05, ## *p* < 0.01, * vs. EXE group, * *p* < 0.05, ** *p* < 0.01, *** *p* < 0.001.

**Table 1 nutrients-16-02831-t001:** The primer sequences are listed. GAPDH, Glyceraldehyde 3-phosphate dehydrogenase; G6Pase, Glucose-6-phosphatase; PC, Pyruvate carboxylase; PCK1, Phosphoenolpyruvate carboxykinase 1; FBP1, Fructose bisphosphatase 1; GK, Glucokinase; GYS2, Glycogen synthase 2; MCT1, Monocarboxylate transporter 1; MCT4, Monocarboxylate transporter 4; HK2, Hexokinase 2; GYS1, Glycogen synthase 1; PGM1, Phosphoglucomutase 1; PDHA1, Pyruvate dehydrogenase E1 alpha 1.

Tissue.	Primer Name	Sequence of Primers (5′–3′)
Liver	GAPDH–F	TGG CCT CCA AGG AGT AAG AAA C
	GAPDH–R	GGG ATA GGG CCT CTC TTG CT
	G6Pase–F	GCC TCC GGA AGT ATT GTC TCA T
	G6Pase–R	CAC CCC TAG CCC TTT TAG TAG CA
	PC–F	GCC CTA TGT TGC CCA CAA CT
	PC–R	GAA CGG GAT GTT CGG GAT AA
	PCK1–F	TGT TCG GGC GGA TTG AAG
	PCK1–R	TCA GGT TCA AGG CGT TTT CC
	FBP1–F	TCT GCA CCG CGA TCA AAG
	FBP1–R	TGG TTG AGC CAG CGA TAC C
	GK–F	TGT GAG GTC GGC ATG ATT GT
	GK–R	CCT TCC ACC AGC TCC ACA TT
	GYS2–F	GAG ACA GTC TTT GCC TCC TGT GA
	GYS2–R	CCT TGA CTC TGT CTG CAC GAT T
	MCT1–F	GTG ACC ATT GTG GAA TGC TG
	MCT1–R	CTC CGC TTT CTG TTC TTT GG
	MCT4–F	CAAAGTGGATCTGCGGTGAA
	MCT4–R	GGCTGGGTCCCTGGTTTAG
Muscle	GAPDH–F	TGG CCT CCA AGG AGT AAG AAA C
	GAPDH–R	GGG ATA GGG CCT CTC TTG CT
	HK2–F	GGG AAG AAG AGA GAC TCG GAA TC
	HK2–R	CAT CCC TGC CTC GCA TAC A
	GYS1–F	CCA GCA CTC GGT AGG TAG AGG TA
	GYS1–R	GTG TCT CAT GTT GCC CAG TTT G
	PGM1–F	GAC GGC CGC TTC TAC ATG A
	PGM1–R	CCA ATA ACC AGG CGA CCA AT
	PDHA1–F	GTG ACC TTC ATC GGC TAG AAG AG
	PDHA1–R	GCA CAG TCT GCA TCA TCC TGT AG
	MCT1–F	GTG ACC ATT GTG GAA TGC TG
	MCT1–R	CTC CGC TTT CTG TTC TTT GG
	MCT4–F	CAAAGTGGATCTGCGGTGAA
	MCT4–R	GGCTGGGTCCCTGGTTTAG

## Data Availability

Data supporting the findings of this study are available from the corresponding author upon request.

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
