# Peer review of "Exogenous Lactate Treatment Immediately after Exercise Promotes Glycogen Recovery in Type-II Muscle in Mice"

_nutrients, 2024, doi:10.3390/nu16172831_

Round 1
Reviewer 1 Report
Comments and Suggestions for Authors
The current manuscript performed an animal study to determine the effects of lactate supplementation after exercise in glycogen metabolism. The authors use a mice model with 8 animals per group and 8 groups including a control one. Even though the authors did not specify how long the mice ran, the rest of the study was well designed including lactate and saline supplementation and multiple time points (1, 3 and 5 hours after exercise). The main results confirmed that lactate supplementation could improve gluconeogenesis in the muscles but not in the liver. This reviewer is very enthusiastic about this manuscript and commends the authors for such elegant study. Even though I do not see major issues to be addressed, the manuscript lacks some consistency that would improve readability.
1. Introduction, lines 50 to 52, it is not very clear if these “studies” are animal or human studies. It would be appropriate to state what type of studies these are.
2. Material and Methods, Experimental design, lines 83-85, it is not clear for how long the mice ran. The figure shows 1 hour, but it should be included in the description of the protocol.
3. Please delete line 89. The figure has its own captions.
4. The font of the figure is too small. You can barely see it. I know that this publication will be accessible in pdf and readers can zoom in the page; however, there are still readers that like printing papers and they won’t be able to read anything.
5. Material and Methods, Experimental design, lines 92-99 appear to be the same as the previous paragraph. Please do not replicate content.
6. Material and Methods, Blood analysis, lines 103-104, you measured triglycerides but you did not report them. Either you delete this sentence from the methods section, or you show the results and then discuss how they are connected to the overall study.
7. Material and Methods, Glycogen Concentration Analysis, line 107, what’s the rationale to use the plantaris muscle? Why did you not use the soleus or quadriceps? Please add a rationale.
8. Results, line 143, were all data normally distributed? Please add a statement about it.
9. Results section includes too many sentences that could go in the Discussion. Please remember that the results section is to state the results of the experiments NOT to explain them. Only facts and numbers. In general, the first and last sentence of each subheading should be revised. For example, lines 195-200; 205-207; 214-216; 237-239, and so forth.
10. All subheadings in the Results section should be broader. For example, 3.1 Blood Lactate and Glucose levels; 3.2 Muscle Glycogen; and so forth. You can use these subheadings in the discussion to direct the readers’ attention.
11. Results, 3.1, line 149, please consider saying that “there was a trend of higher BLa at 3-hr”.
12. Figure 2, 3 and 5, please consider showing data from CON and EXE as a 0-hr, similar than Figures 4 and 6. They could stay as they are, but it was confusing to see that what happened before hour 1 was plotted after hour 5 (I hope it makes sense).
13. Lines 155, 177, 217, 240, and 267 should be deleted. All figures have their own captions.
14. Results, 3.2 and 3.4, please be consistent with the decimal figures. Sometimes you use 2 and sometimes 3 for the same units. For same units, the same number of figures after the decimal point should be in the average and SDs.
15. Results 3.3, 3.5, and 3.6, please be consistent in how you show your results. In 3.3 you did not include the average and SD with “expression level/GAPDH” statement. Either you use it in all gene results or you do not use it in all gene results. I think 3.3 is the cleanest.
16. Results about MCT1 (line 213) do not show in the figure. In fact, the figure says it was significant, but the text says it was not. Please clarify.
17. Figures 4 and 6, please increase the font of # and *. You can barely see them.
18. Overall, the discussion needs more details. For example, authors should consider a mentioning the authors and a brief description of the other studies referred in the discussion. Statements such as “unlike that in the chronic exercise experiment” *lines 315-317, or “a previous study” (line 330), etc.
19. In addition, the discussion would improve if the authors use subheadings, like the suggestion above (#10).
20. Authors should be more cautious with their language. The results of the current study are not definitive to say “La CAN be absorbed…” La could be absorbed. The study was not design to test the hypothesis that La is well absorbed or not.
21. Lines 324-326, it is not clears what the authors mean here. If the findings “underscore” the need of closer examination, why are you presented gene expression? Please re-write the sentence.
22. Line 393, “in” the liver and not “on” the liver.
Comments on the Quality of English LanguageNo more comments.
Author Response
Thank you for taking the time to review our manuscript. We sincerely appreciate your valuable comments and the contribution they have made to improving our work.
"Please see the attachment."
There were two attachments, but only one could be uploaded, so I have combined them into a single file.
- Pages 1-16 contain the revised manuscript.
- Pages 17-29 contain our responses to the comments.

Reviewer 2 Report
Comments and Suggestions for Authors
Thank you for the opportunity to review the article with the title - Exogenous lactate treatment immediately after exercise promotes glycogen recovery in type-II muscle in mice. The article is well structured, but to improve it we recommend the following:
Introduction:
- To detail the new aspects of the present study by referring to previous studies on the same topic.
- To add the hypotheses of the study.
Method - To add the periodization of the study, when the study was started, the time stages of the study and when the study was completed.
Discussions - to add the practical implications of the study focusing on the effects on the sports training process.
Conclusions - to expand the conclusions in relation to the relevant results identified.
Author Response
Thank you for taking the time to review our manuscript. We sincerely appreciate your valuable comments and the contribution they have made to improving our work.
There were two attachments, but only one could be uploaded, so I have combined them into a single file.
Pages 1-16 contain the revised manuscript.
Pages 17-19 contain our responses to the comments.

Round 2
Reviewer 1 Report
Comments and Suggestions for Authors
I commend the authors for their thorough review of their manuscript, which has improve its readability. There are still a few details that need to be addressed.
1. Line 52 says “rates”, I think it is “rats”
2. I understand that in the methods section, line 154-155, the authors stated that they will analyze the data for normality. However, in the results, they still have to state if the data was or not normally distributed.
3. Some subheadings in the Results section are still too specific. For example, 3.3, 3.5, and 3.6. Please delete “Effects of Exogenous Lactate Treatment on” in all of them.
4. The discussion still needs more details. For example, instead of saying “ a previous study”, you should acknowledge the authors from that previous study (Cori et al found that…..). This should be done for each of the studies discussed (refs. #21, 30, 31, 36)
5. Reference 21, Carl Cori et al, looks incomplete.
Author Response
Point-by-point response to Comments and Suggestions for Authors
General Comments: I commend the authors for their thorough review of their manuscript, which has improve its readability. There are still a few details that need to be addressed.
Response : Thank you sincerely for your significant help in revising my manuscript. I carefully reviewed the five suggested revisions you provided and took the opportunity to identify and correct additional errors and typos. Below are my responses to the five comments you mentioned:
Dear [Reviewer comments - 1],
Comments 1: Line 52 says “rates”, I think it is “rats”
Response 1 : Thank you so much for your careful review and kind feedback. Thanks to your detailed comments, I was able to correct the typos I had missed. I have since thoroughly checked all my sentences for any additional errors and made the necessary revisions. Your valuable input has greatly enhanced the quality of the document.
Line in the revised manuscript | Before | After |
Line 52 | Rates | Rats |
Dear [Reviewer comments - 2],
Comments 2: I understand that in the methods section, line 154-155, the authors stated that they will analyze the data for normality. However, in the results, they still have to state if the data was or not normally distributed.
Response 2 : In response to the reviewer's request, I have added the following phrases to each result to enhance the natural flow of the explanations. Thank you for your detailed guidance.
Line in the revised manuscript | Before | After |
Line 172 to 175, 196 to 199, 234 to 237, 259 to 262, 283 to 285, 318 to 321 |
--- |
The normality of the (result) analysis data was assessed using the Shapiro-Wilk or Kolmogorov-Smirnov test. The results indicated that the data were normally distributed (p > 0.05). Therefore, parametric tests were applied in further analyses. |
Dear [Reviewer comments - 3],
Comments 3 : Some subheadings in the Results section are still too specific. For example, 3.3, 3.5, and 3.6. Please delete “Effects of Exogenous Lactate Treatment on” in all of them.
Response 3 : In response to the reviewer's request, I have revised the wording of the results subheading. Thank you for the detailed information.
Line in the revised manuscript | Before | After |
Line 205 |
3.3. Effect of Exogenous Lactate Treatment on Intramuscular Glycogen Synthase and MCT-1,4 Gene Expression |
3.3. Glycogen Synthase and MCT-1,4 Gene Expression in muscle |
Line 265 |
3.5. Effect of Exogenous Lactate Treatment on Liver Gluconeogenesis Gene Expression |
3.5. Gluconeogenesis Gene Expression in Liver |
Line 299 |
3.6. Effect of Exogenous Lactate Treatment on Liver Glycogen Synthesis and MCT-1,4 Gene Expression |
3.6. Glycogen Synthesis and MCT-1,4 Gene Expression in liver |
Dear [Reviewer comments - 4],
Comments 4 : The discussion still needs more details. For example, instead of saying “ a previous study”, you should acknowledge the authors from that previous study (Cori et al found that…..). This should be done for each of the studies discussed (refs. #21, 30, 31, 36)
Response 4 : In response to the reviewer's request, I have revised the manuscript to provide a more detailed explanation than the word "previous research." Thank you for the detailed explanation.
Line in the revised manuscript | Before | After |
Line 47, Ref #21 | Lactate produced during exercise is converted back to glucose in the liver via the Cori cycle or intracellular movement of lactate [18,21]. | Cori et al found that, revealing that lactate produced in muscles during anaerobic glycolysis is transported to the liver, where it is converted back into glucose. This glucose is then reused by the muscles as an energy source, a process now known as the "Cori cycle" [21]. |
Line 99, Ref#30 | "On reviewing the cited paper, I realized it did not address the comparison of lactate concentrations as initially thought. I have since removed the reference, ensuring that the content accurately reflects the scope and focus of the research. Thank you for your insightful feedback, which has helped to maintain the integrity and clarity of the manuscript." | immediately after the treadmill ex-ercise at 3 g/kg (sodium L-lactate, L7022, Sigma) [26–29]. |
Line 350, Ref#31 | A review article on muscle glycogen and body water found a correlation between muscle glycogen content and weight gain in humans but noted that the results of animal studies were inconsistent. For example, while a correlation was observed in the liver, studies on muscle showed mixed results [31,32]. | A review article on muscle glycogen and body water identified a correlation between muscle glycogen content and weight gain in humans [31]. However, Sherman et al. found no consistent relationship between water volume and glycogen content in rodent skeletal muscle, highlighting potential species-specific differences in the relationship between glycogen storage and water retention [32]. |
Line 359, Ref#36 | Additionally, an increase in the expression of HK2, also known as the "guardian of mitochondria" [35], has been confirmed to enhance exercise performance[36]. | Additionally, an increase in the expression of HK2, often referred to as the "guardian of mitochondria” [35], has been linked to improved muscle endurance. Fueger et al. observed that higher levels of hexokinase (HK) protein content were specifically associated with enhanced muscle endurance in their study [36]. |
Dear [Reviewer comments - 5],
Comments 5 : Reference 21, Carl Cori et al, looks incomplete.
Response 5 : In response to the reviewer's request, I have updated all of the references to align with the Nutrients format. I appreciate your guidance in ensuring the manuscript meets the required standards. Please let me know if there are any further adjustments needed.
ex;
Line in the revised manuscript | Before | After |
Line 536 | 21. Carl Cori, B.F.; Cori, G.T. GLYCOGEN FORMATION IN THE LIVER FROM D-AND Z-LACTIC ACID; | Cori, C.F. and Cori, G.T., 1929. Glycogen formation in the liver from d-and l-lactic acid. Journal of Biological Chemistry, 81(2), pp.389-403. |
